# Primary Barriers of Adherence to a Structured Nutritional Intervention in Patients with Dyslipidemia

**DOI:** 10.3390/nu13061744

**Published:** 2021-05-21

**Authors:** Fabiola Mabel Del Razo-Olvera, Angélica J. Martin-Vences, Griselda X. Brito-Córdova, Daniel Elías-López, María Victoria Landa-Anell, Marco Antonio Melgarejo-Hernández, Ivette Cruz-Bautista, Iliana Manjarrez-Martínez, Donají Verónica Gómez-Velasco, Carlos Alberto Aguilar-Salinas

**Affiliations:** 1Research Unit of Metabolic Diseases, Instituto Nacional de Ciencias Médicas y Nutrición Salvador Zubirán (INCMNSZ), Mexico City 14080, Mexico; fabiola.delrazoo@incmnsz.mx (F.M.D.R.-O.); drdanielelias@outlook.com (D.E.-L.); ivette.cb27@gmail.com (I.C.-B.); donajivgv@gmail.com (D.V.G.-V.); 2Deparment of Endocrinology and Metabolism, Instituto Nacional de Ciencias Médicas y Nutrición Salvador Zubirán (INCMNSZ), Mexico City 14080, Mexico; grisbrito@hotmail.com (G.X.B.-C.); manut82004@yahoo.com.mx (M.A.M.-H.); manjarrezilian@yahoo.com.mx (I.M.-M.); 3Escuela de Altos Estudios en Salud, Universidad La Salle, Mexico City 14010, Mexico; amartinan.v@gmail.com; 4Centro de Atención Integral del Paciente con Diabetes, Instituto Nacional de Ciencias Médicas y Nutrición Salvador Zubirán (INCMNSZ), Mexico City 14080, Mexico; nutlanda@gmail.com; 5Division of Nutrition, Instituto Nacional de Ciencias Medicas y Nutricion Salvador Zubirán (INCMNSZ), Vasco de Quiroga #15, Tlalpan, Mexico City 14080, Mexico

**Keywords:** dietary adherence strategies, blood lipids, macronutrients

## Abstract

Purpose: To describe the primary barriers to adequately adhering to a structured nutritional intervention. Patients and methods: A total of 106 participants diagnosed with dyslipidemia and without a medical nutrition therapeutic plan were included in this two-year study conducted at the INCMNSZ dyslipidemia clinic in Mexico City. All patients were treated with the same structured strategies, including three face-to-face visits and two telephone follow-up visits. Diet plan adherence was evaluated at each site visit through a 3-day or 24-h food recall. Results: Barriers to adhere to the nutritional intervention were: lack of time to prepare their meals (23%), eating outside the home (19%), unwillingness to change dietary patterns (14%), and lack of information about a correct diet for dyslipidemias (14%). All barriers decreased significantly at the end of the intervention. Female gender, current smoking, and following a plan of more than 1500 kcal (R^2^ = 0.18 and *p*-value = 0.004) were associated with good diet adherence. Participants showed good levels of adherence to total caloric intake at visit 2 and 3, reporting 104.7% and 95.4%, respectively. Adherence to macronutrient intake varied from 65.1% to 126%, with difficulties in adhering to recommended carbohydrate and fat consumption being more notable. Conclusion: The study findings confirm that a structured nutritional intervention is effective in reducing barriers and improving dietary adherence and metabolic control in patients with dyslipidemias. Health providers must identify barriers to adherence early on to design interventions that reduce these barriers and improve adherence.

## 1. Introduction

Cardiovascular diseases (CVDs) are the leading cause of death worldwide [1,2], including Mexico [3], where CVD-related deaths occur even before age 50 [4]. Numerous metabolic and behavioral risk factors have been associated with CVDs, including dyslipidemias; these disorders are a silent risk factor and play a crucial role in disease progression [5]. Diet is also a determinant in CVD and is the main behavioral factor directly contributing to CVD’s disease burden [6]. Lifestyle modifications, therefore, play a fundamental role in improving patients’ lipid profiles and in maintaining optimal cardiovascular health [7].

Medical Nutritional Therapy (MNT) is considered the cornerstone of dyslipidemia treatment [8]. Scientific evidence supports referring patients with dyslipidemia to a certified nutritionist for proper management [9], as MNT is effective in preventing and reducing cardiovascular risk in these patients [10] and additionally offers the best cost–benefit ratio [11]. Yet, despite its advantages, MNT is the treatment with the largest long-term failure percentage. Therefore, improving and maintaining long-term patient adherence is essential for preventing and controlling the disease [12,13]. However, the barrier to adherence to MNT and the proposals to resolve these barriers have not been sufficiently studied over time, even though the prevalence of the barriers has increased [14].

This study set out to describe the main barriers to achieve adequate MNT adherence and evaluate the effect of a structured nutrition intervention on MNT adherence and lipid profile in patients with dyslipidemia.

## 2. Materials and Methods

### 2.1. Participants and Procedures

From 2016 to 2018, all first-time patients with a documented diagnosis of dyslipidemia and without an MNT plan previously prescribed by another department of dietetics, either internal or external to the Instituto Nacional de Ciencias Médicas y Nutrición Salvador Zubirán (INCMNSZ), a level three hospital in Mexico City, were invited to participate in this study. A total of 106 participants diagnosed with dyslipidemia (61 women and 45 men) were included. Inclusion criteria included being literate, above the age of 18, receiving treatment at the INCMNSZ dyslipidemia clinic, and having a documented dyslipidemia diagnosis. Patients had to voluntarily agree to take part in this study and sign written informed consent before participating. Exclusion criteria included patients who had presented comorbidities such as acute renal failure, nephrotic syndrome, or any other clinical condition that—according to the researchers—required different nutritional therapies than those used in the dyslipidemia clinic. All patients were treated with the same structured intervention (Table 1). Barriers to adherence were collected at each visit through a questionnaire. The participants were asked to select one of the barriers described there or to write another barrier if none of those mentioned was the main issue for them. Three months prior to the start of the study, three registered dietitians, external to the research, determined the most common barriers through in-person interviews and a literature review.

Patients received specific advice and color printed material to help them overcome each barrier detected. During each visit, we used specific materials to solve adherence barriers such as low cost menu examples and answers to the following questions: “What can I do if I eat away from home?”, “What can I do if I don’t have enough time to prepare my meals?”, “What kind of food increases cholesterol and triglycerides?”, “How do you read food labels?” among others. Specific advice was provided according to the barriers detected, which was done in accordance with the context of the patient. Furthermore, we also tried to provide tools to eliminate or reduce the detected barrier.

We implemented two different plans depending on the needs of the patient. (1) Participants received a complete food exchange system made up of 12 black and white sheets, showing the different food groups and quantities per food serving; the serving numbers were specified for each mealtime. (2) Participants were informed using a traffic light system, with red for foods to avoid, yellow for foods to consume in moderation, and green for foods that can be eaten without restriction. Instead of specifying food servings for each mealtime, the plan only prescribed the total servings of each group of foods during the day. Each medical nutrition plan was calculated using the nutritional requirements and lipid therapeutic goals of each patient. We used the Harris–Benedict formula, adjusted by the activity factor, to prescribe the energy content of the diet [15]. General nutritional and physical activity recommendations were given as an integral part of the non-pharmacological treatment as stated by “The National Cholesterol Education Program Adult Treatment Panel IV” (ATP IV) clinical guidelines. Within each nutritional plan, the patients were instructed to avoid the consumption of foods with saturated and trans-fat such as fried foods, margarines, organ meats, and industrialized bread, among others. Apart from calculating the requirements of each patient, we also used data from the initial interview such as religion, eating habits, type of exercise, and the 24-h food recall to prescribe the nutritional plan and avoid drastic changes in diet. The diets were made up of 40 to 55% carbohydrates, 15 to 20% protein, 30 to 40% fat, and 25 to 35 gr of fiber/day. The meal planning system was distributed in three main meals. If snacks were necessary, they were prescribed on an individual basis.

Adherence to the medical nutrition therapy was evaluated through a 3-day food recall because it is a cost-effective tool, although it may have limitations, such as those described by Subar et al. [16]. Unfortunately, we did not measure dietary biomarkers to calibrate nutrient consumption self-reports. At each visit, the patients were given a 3-day food recall (with the instruction to register two weekdays and one weekend day). At subsequent visits, the 3-day food recall was used to measure patients’ adherence to the nutritional plan; when this was not available, which was the case for 14.15% of the participants, we used a 24-h food recall. The evaluation of the records was done using the Food Processor ^®^ program; a micronutrient analysis was not performed. Due to the lack of evidence that indicates an ideal cut-off point to establish adequate adherence to nutritional treatment, this study established the following categories: patients were classified as having “good adherence” when the concordance of macronutrients (carbohydrates, proteins, and lipids) was not above 110% nor below 80% of the total grams prescribed. Concordance of adherence to energy intake was defined as a total caloric intake between 80% to 110% of the prescribed and “bad adherence” when this percentage was below 80% or above 110%.

Adherence = (Energy (kcal) or macronutrient (g) consumed/Energy or macronutrient prescribed) × 100

During all their visits, all participants were advised on specific dietary and lifestyle recommendations related to dyslipidemia diagnosis. The visit start and stop times were recorded to measure how long the session lasted. A registered dietitian and expert in dyslipidemia management supervised and standardized the patients’ care plans. All information about the structured intervention is described in Table 1. Participants visited the dyslipidemia clinic three times throughout the entire study period; on-site visits were scheduled every six months. The intervention was reinforced through telephone visits. Additionally, we asked about the adherence to the nutritional plan using a Likert scale from 0 to 100 (zero being the lowest and 100 being the highest rating).

The electronic medical record was used to collect data on the patients’ clinical history. All participants were given an initial interview to obtain the personal data not found in the electronic patient chart, such as family history, personal history, pathological background, comorbidities, and current pharmacological treatments.

Anthropometric measurements (weight, waist, and hip circumference) and dietary evaluations were performed at each visit, while height was only measured at the first visit. Body weight was taken with a calibrated digital scale, with an accuracy of 100 g. Height was measured using a stadiometer with a fixed vertical backboard and adjustable headboard with an accuracy of 0.5 cm. These measurements were taken with individuals wearing light clothing and barefoot, and without metallic props or objects on the head or in the pockets. Body Mass Index (BMI) was calculated as the weight in kilograms divided by the square of the height in meters.

Biochemical laboratory tests, including a lipid profile, were done 8 to 15 days before the visit and recorded on the day of the visit. All the biochemical tests were carried out in the central laboratory of the INCMNSZ. A data emptying sheet was used to record anthropometric and dietary variables (24-h and 3-day food recalls). At each visit, patients were asked about the kind and duration of the physical activity or exercise they performed per day. All subjects carried out their conventional medical treatment.

### 2.2. Ethics

The study protocol was approved by the INCMNSZ’s Research Ethics Committee, with reference number 1511.

### 2.3. Statistical Analysis

As a first step, we used a Kolmogorov–Smirnov test to determine the distribution of the data. Participant characteristics then were described using mean ± SD or median and interquartile range. Differences between visits were analyzed by repeated measures ANOVA and Wilcoxon tests for continuous variables. A chi-square (X^2^) test was used to test for significant differences in categorical variables’ frequency distributions. The quantitative relationship between the variables and adherence to MNT was analyzed using a logistic regression model, and the results are presented through odds ratio values (OR), with a confidence interval of 95%. To identify potential variables for multivariate analysis, we used two criteria: (1) “theoretical variables” based on the literature and (2) a previous statistically significant correlation (*p* < 0.05). The variables were included in the multivariate model through the “Intro” method, and variables with a significant *p*-value (<0.05) or variables with biological plausibility remained in the final model. The analyses were performed using the IBM Statistical Package for Social Sciences (IBM SPSS), version 21.0 (SPSS Inc., Chicago, IL, USA).

## 3. Results

Out of 183 possible participants, 43 did not meet inclusion criteria, 30 were not included for personal reasons (such as not having enough time to complete the study), 2 were excluded for refusing to sign informed consent, and 2 were lost to follow-up. Full results were obtained from 106 patients with dyslipidemia (Figure 1). Of them, 61 were women and 45 were men. Participants completed a total of five visits; they visited the dyslipidemia clinic three times—once at the beginning of the study and twice during the follow-up—with an average period between visits of 6.7 ± 4.4 months. Participants also received two phone call visits to evaluate adherence to the nutritional plan between visits 1 and 2 and between visits 2 and 3.

The average participant age was 46.8 ± 12.5 years. The most common dyslipidemias found in participants were primary dyslipidemias, such as familial combined hyperlipidemia (FCHL). Medium and high-intensity statins were prescribed in 89.6% of patients at doses of 10, 20, or 40 mg. Fibrates were the second-most prescribed treatment, indicated by 50.5% of the patients, and 24 (22.6%) participants used a different kind of lipid-lowering drug. Of the participants, only 11 (10.4%) were not receiving any treatment because they were under adequate metabolic control. All sociodemographic variables, such as years of schooling, tobacco use, alcohol intake, and family history of dyslipidemias, are described in Table 2.

### 3.1. Barriers

Upon exploration of the barriers to adherence to MNT, patients received specific educational advice and educational tools (described above) to help overcome each detected barrier. There was a significant change in their prevalence: while 14% of the 106 participants reported “I don’t need/want to make changes to my diet” as a barrier to adherence for MNT, this number decreased to 13% at V2 and to 0% at V3. This barrier was effectively reduced by 100% (*p* < 0.001) at the end of the study. The barrier “Lack of information about the correct diet for dyslipidemias” was reported by 13% of participants at baseline, which then decreased to 0% at visit 2 and 0.5% at visit 3; this barrier was reduced 92.86% (*p* < 0.001) at the end of the study. In the same vein, 19% of the patients reported “I eat away from home most of the time” as a barrier—14% at V2 and 0% at V3; this barrier was reduced by 90% at the end of the study (*p* < 0.001). While 11% of the patients reported “Economic situation (“Being on a diet is very expensive”) as a barrier at baseline, it decreased to 7% at V2 and 0.5% at V3; the barrier was reduced 91.67% (*p* < 0.001) at the end of the study. The barrier “Lack of time to prepare my meals” was reported in 13% of participants at baseline, increasing slightly to 14% at V2, and then decreasing to 4% at V3; this barrier was reduced by 83.33% at the end of the study (*p* < 0.001). Table 3 shows the analyzed barriers. At the beginning of the study, one-fifth (20%) of the sample size did not identify any barriers to follow the dietary instructions; however, at the end of the study, 93% of participants declared the absence of barriers (*p* < 0.001) for following their nutritional plan.

### 3.2. Adherence and Dietary Patterns

Adherence to MNT was evaluated through 3-day food recalls and was also self-reported during phone call visits follow-up, using a Likert scale ranging from 0 to 100. We noticed that patients reported less adherence to the MNT to the registered dietitian, in comparison to the evaluation of 3-day food recalls. Patients tend to be critical; therefore, they might feel that they “deserve” a poor grade when not following nutritional recommendations. However, when we reviewed the 3-day food recalls, we realized that participant adherence was higher than what they were reporting. Reported adherence was 60% and 70% during visit 2 and visit 3, respectively (*p* < 0.001).

Regarding total energy intake, the patients showed good adherence between V1 and V2 (104.7%) and V2 and V3 (95.4%), and although the percentage of adherence decreased, it was not significant. Likewise, the participants presented poor adherence to carbohydrate intake recommendations, showing excessive intake between V1 and V2 (126.8%) and between V2 and V3 (112.4%); even though adherence improved, it was considered poor. Adherence to protein intake was considered good and decreased during the study, although not significantly: 104.8% between V1 and V2 and 96.6% between V2 and V3. Fat intake adherence was good between V1 to V2 (100.3%); however, it declined significantly to 64.1% (*p* < 0.001), during V3, and was, therefore, considered poor.

In terms of dietary patterns, participants presented a reduction of 364 Kcal in energy intake (*p* < 0.001), which is considered significant. At the beginning of the study, we observed a moderately high carbohydrate intake of 266.6 g. It decreased significantly to 193.9 g with the intervention (*p* < 0.001). There was also a reduction in protein and fat intake during the study (*p* < 0.001). This information is described in Table 4. Although the fat intake reduction was higher than expected with the intervention, we observed that participants believed that fat consumption worsened their health.

We explored all factors that could have possibly been related to adhering to the plan in 80–110% at V3. Through a bivariate analysis, we obtained five variables with statistically significant (*p* < 0.05) correlation with good adherence: sex, smoking, prescribed Kcal, barriers, and occupation (data not shown) to build a logistic regression model. We found that being a woman, following a nutritional plan with more than 1500 kcal, or being a current smoker were factors associated with better adherence. By incorporating occupation as a predictor variable in the model, this variable loses statistical significance and does not improve the performance of the model; therefore, occupation was excluded. We included the absence of reported barriers at V3; even though this variable does not have a statistically significant *p*-value, it tends to be significant (*p* = 0.008). Hence, 18% of the variability in adherence to the MNT in this study can be explained by these factors (Table 5).

### 3.3. Metabolic Control/Lipid Profile Changes

In terms of clinical and anthropometric variables, between V1 and V3, there was a slight weight loss. However, this loss was not significant at the end of the intervention as operationalized by BMI. There was a significant decrease in waist circumference by 1.8 cm (*p* < 0.001). Blood pressure levels remained within normal range throughout the study. Blood lipid levels improved significantly in participants with MNT, who achieved better blood lipid control during the follow-up period. Thus, triglycerides and total cholesterol levels were reduced significantly from 215.2 mg/dL to 165.5 mg/dL and 215.2 to 198.4 mg/dL, respectively (*p <* 0.001). At the end of the study, adherence to the triglyceride target goal (<150 mg/dL) reached significance, as it was achieved by 41.5% of the participants (*p* < 0.001). The adherence to the total cholesterol goal (<200 mg/dL) also reached significance, as it was achieved by 51% of the participants (*p* < 0.001). All differences before and after the structured intervention with MNT are described in Table 6.

## 4. Discussion

This study sought to describe the main barriers over time that patients with dyslipidemia face when attempting to adhere—in the short term—to MNT and the effect that this therapy has on their lipid profile.

### 4.1. Barriers to Adherence

Poor adherence to MNT is related to the presence of barriers that prevent patients from following a healthy diet [17]. Therefore, health providers can identify early barriers to follow a nutritional plan and design structured interventions to reduce those barriers [18]. Landa et al. designed a survey to detect barriers to adherence during a nutritional intervention. By the end of their study, 42.8% of their participants reported a lack of barriers after a two-year follow-up period [18]. Our results concur with Landa et al. At the beginning of our study, 20% of the patients did not recognize any barriers to following dietary recommendations. However, by the end of the study, this percentage rose significantly to 93%. Additionally, our study analysis reported that the main barrier to adherence was “Lack of time to prepare my meals.” The prevalence for this barrier fell from 23% to 4% between the study’s beginning and end. Moreover, at V1, 14% of the patients reported having the barrier “I don’t need/want to make changes to my diet,” but it dropped significantly to 0% by V3; this barrier was reduced by 100% at the end of the follow-up period. Indeed, this barrier to adherence has to do with willpower and motivation [19]. Therefore, it can be implied that patients felt motivated at the end of the follow-up period. Evidence has shown that motivation is required to assure adherence to MNT. Hence, recommendations to improve adherence should be simple, emphasizing the teaching and motivation methods as motivation levels impact the learning process [20,21]. Our results show that a structured MNT intervention, early detection of barriers to adherence, and educational advice are effective in motivating the patient to follow this kind of therapy.

### 4.2. Adherence

Previous research has explored how nutritional interventions can improve adherence and the metabolic profile of patients with chronic diseases. In a systematic review, Desroches et al. found good adherence to MNT in studies using multiple strategies; they also found educational tools and feedback as promising interventions for managing chronic diseases. Still, the same authors were unable to isolate the effect of each individual intervention itself [22]. In our study, it was also found that educational tools and feedback helped to improve overall adherence between V1 and V2; however, it decreased from V2 to V3. Additionally, a significant increase in self-reported adherence is found between V2 and V3. We observed that the adherence reported by the patient in the 3-day food recall is higher than that reported in the telephone reports (self-reported adherence), which suggest a bias.

Our results concur with those reported by Kendall et al., who compared two different strategies (exchange food vs. a diet guide) to assess adherence to MNT. In their study, they did not find any difference between the use of one strategy or the other. In addition, they found a significant reduction in caloric and lipid intake. However, their results do not show any significant differences in the lipid profile, weight loss, or blood pressure levels [23]. In our study, we observed a significant decrease in energy and macronutrient intake, and also there were significant differences in body weight, but not in BMI and blood pressure. However, contrary to Kendall et al., we observed a significant reduction in waist circumference, which is essential to decrease cardiovascular risk [9,24,25]. In our study, good adherence was observed both in the group with the complete food exchange system, as well as in the group that followed the simplified plan (traffic light system); therefore, it cannot be asseverated that one plan is better than the other.

Long-term adherence is a determining factor for obtaining adequate disease prevention and management [9,26]. In our study, participants showed adequate levels of adherence between V1 and V2 in terms of energy, protein, and fat intake. However, adherence to fat intake declined statistically in the last visit. These outcomes concur with Henkin et al.’s results, who found a favorable initial response to diet in patients with dyslipidemia but adherence issues by the study’s end. The authors conclude that long-term adherence to MNT in patients with dyslipidemia is difficult to achieve. The authors also suggest continuous patient support to improve adherence [27]. In our study, we found that early detection of barriers, along with telephone follow-up and simple educational tools, is effective in motivating the patients. However, this is only helpful for obtaining MNT adherence in the short term.

Our results also show that being a woman, eating a diet with more than 1500 Kcal, and being a current smoker are factors related to better adherence to MNT. In addition, “To have or not to have barriers” shows a trend towards statistical significance, in relation to a better adherence. However, the results raise questions on how being a current smoker can influence adherence to MNT and how barriers to adherence can be lowered while at the same time adherence to MNT declines. The results of Chao et al. might answer these questions, as they studied how demographic, clinical, and psychological factors are related to smoking status and specific food cravings. Their study mentions how smoking is an appetite suppressant, and people perceive it as a coadjutant for weight management. The authors conclude that food cravings and current smoking may be related to people with stress and depressive symptoms. Their results may explain how smoking is related to better adherence to MNT; however, we shall not misunderstand the result, as smoking is a cardiovascular risk and an important cause of mortality. On the other hand, Chao et al. also found that nicotine dependence is related to an increase in food cravings. Current smokers were reported to crave more foods rich in fats [28]. Surprisingly, this could justify, in our results, how adherence to fat intake declined in V3, even in the absence of barriers to adherence to MNT. Furthermore, Diekman et al. conducted a survey in 16 countries that included 6426 participants. In this study, the authors pointed out that people have a lack of knowledge regarding dietary fats. In fact, 90% of the people surveyed associated dietary fat with negative health effects. Therefore, when patients do not know the importance of the quality of fats in the diet, they choose to limit its intake. The authors concluded that it is necessary to educate both health workers and the population regarding the types of fats and the importance of these in their health [29]. Therefore, it is crucial to educate patients and also evaluate behavioral, emotional, and stress factors to achieve better MNT adherence in the long term.

### 4.3. Metabolic Control/Lipid Profile Changes

MNT has shown to be effective in managing blood lipids and reducing cardiovascular disease risk [9,25]. In our study, we show a significant improvement in blood lipid levels, with significant decreases in triglycerides and total cholesterol levels during MNT interventions. The literature surrounding MNT’s benefits in dyslipidemias is compelling, describing that a reduction in saturated fat intake and an increase in dietary polyunsaturated fatty acids was associated with reductions of plasma total cholesterol and C-LDL [30].

MNT has also been shown to be cost-effective. In a systematic review with a meta-analysis, MNT was clinically effective (even without concurrent hypolipidemic drugs), improving quality of life, dietary intake, and physical activity performance [9]. Additionally, our study found that a structured nutrition intervention can improve the metabolic profile of patients with dyslipidemias. Therefore, we posit that MNT offers the most effective cost–benefit ratio and the greatest clinical benefits to these patients’ metabolic profiles.

The limitations in this study include the number of consultations per patient during the study. Three consultations do not allow evaluating the behavior during an MNT intervention. Likewise, the follow-up time between each consultation is long (6–7 months), even when telephone attention was given to answer questions and reinforce knowledge. Additionally, there was no instrument implemented to measure the impact of the intervention on disease knowledge nor patient motivation. The questionnaire to explore the barriers to adherence to TMN has not been validated in Mexican population, which constitutes an area of opportunity for future research. Furthermore, measuring adherence through a 3-day food recall, filled by the patient, depends on memory, adequate estimation of the portions, and objectivity, which may represent biases when measuring adherence. Finally, we did not offer psychological or behavioral advice to consider stress, anxiety, or depression symptoms in the participants.

To the best of our knowledge, this study of a structured nutrition intervention on MNT is the first to evaluate barriers to adherence in patients with dyslipidemias. Our results provide resounding evidence that it is necessary to identify dyslipidemia patients’ unique needs for implementing tailored interventions. It is necessary to apply psychological evaluations to measure motivation and/or the stages of change in this kind of patients. Rather than just giving an elimination food list, dietitians should detect early barriers to adherence to MNT. Health providers should place more emphasis on nutritional education—not only on how to follow a nutritional plan but also on health risks and benefits. Lastly, dietitians should adjust educational tools and teaching methods to the corresponding level of patient literacy, leaving easy-to-understand messages, to allow the patient to feel motivated and fortify MNT adherence in the long term.

## 5. Conclusions

During a structured intervention of MNT in patients with dyslipidemia, it was found that the main barriers to adherence are related to financial difficulties, lack of time, will-power, and the patient’s education. Finally, early detection of barriers, motivation, and the patient’s education improve adherence to MNT in the short term, improving the lipid profile in patients with dyslipidemia.

## Figures and Tables

**Figure 1 nutrients-13-01744-f001:**
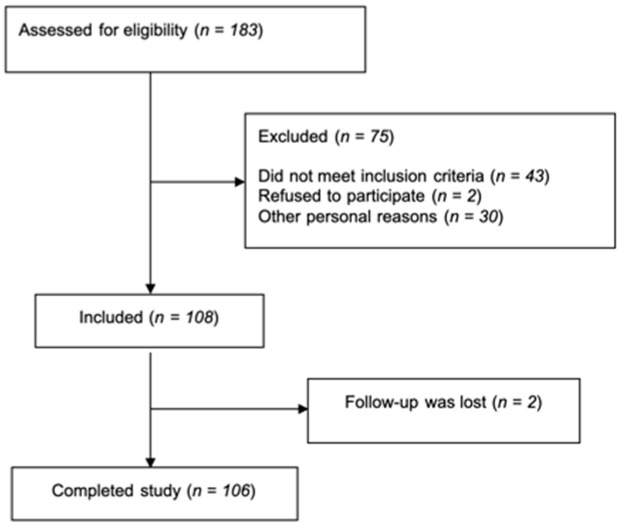
Selection of patients: flow diagram.

**Table 1 nutrients-13-01744-t001:** Strategies to improve Medical Nutrition Therapy (MNT) adherence in a structured nutritional intervention.

	Principal Objective	Material for Visit
Visit 1	Identifying foods that increase triglycerides and cholesterol. To provide a personalized plan based on each participant’s characteristics. To assess barriers to adherence to nutritional treatment.	Color information cards: “Foods that increase triglycerides and foods that increase cholesterol.” Simplified nutritional plan according to traffic light colors or a food exchange list. Color cards with different strategies were provided to solve each of the barriers, i.e., “What can I do if I eat outside my home?” or “How do you read food labels?”
Telephone visit	To assess adherence to diet by self-reports. To solve patient questions and concerns.	Telephone interview: To ask patients to rate adherence themselves on a scale of 0 to 100. Provide advice.
Visit 2	Evaluation of adherence by a 3-day or 24-h food recall. We compared the quantity and quality of food against the recommended portions. Identification of barriers and solutions. Examples of breakfast, lunch, or dinners according to their nutritional plan. Healthy snacks.	A 3-day or 24-h food recall. Color cards with different strategies to solve each of the barriers. Written information. Color cards with different options.
Telephone visit	To assess diet adherence through a self-report. To solve patient questions and concerns.	Telephone interview: To ask patients to rate adherence themselves on a scale of 0 to 100. Provide advice.
Visit 3	Evaluation of adherence by a 3-day or 24-h food recall. We compared the quantity and quality of food against the recommended portions. Identification of barriers and solutions.	A 3-day or 24-h food recall. Color cards with different strategies to solve each of the barriers.

**Table 2 nutrients-13-01744-t002:** Baseline characteristics of the participants.

Characteristic	n (%)
Sex Female Male	61 (57.5%) 45 (42.5%)
Age	46.8 ± 12.5
Years of study	13.5 (9–19)
Current smoker Yes	22 (20.7%)
Current alcohol consumption Yes	41 (38.6%)
Type 2 diabetes Yes	21 (19.8%)
Hypertension Yes	27 (25.4%)
Family history of Dyslipidemia Yes	64 (60.3%)
Type of Dyslipidemia Primary Secondary	59 (55.7%) 47 (44.3%)
**Dyslipidemia Treatment** Statins Fibrates Ezetimibe Niacin Omega 3 Other No treatment	95 (89.6%) 53 (50%) 11 (10.3%) 1 (0.9%) 11 (10.3%) 1 (0.9%) 11 (10.4%)

**Table 3 nutrients-13-01744-t003:** Frequency of barriers to adhering to nutritional treatment.

Barriers to Adhering to Nutritional Treatment	Visit 1 n (%)	Visit 2 n (%)	Visit 3 n (%)	*p*-Value §
Lack of information about a correct diet for dyslipidemias	14 (13)	0 (0)	1 (0.5)	<0.001
Economic situation (“Being on a diet is very expensive”)	12 (11)	7 (7)	1 (0.5)	<0.001
Lack of time to prepare my meals	24 (23)	15 (14)	4 (4)	<0.001
I eat away from home most of the time	20 (19)	19 (18)	2 (2)	<0.001
I don’t need/want to make changes to my diet	15 (14)	14 (13)	0 (0)	<0.001
None	21 (20)	51 (48)	98 (93)	<0.001

§ Chi^2^ test between visit 1 and visit 3.

**Table 4 nutrients-13-01744-t004:** Nutritional treatment adherence analyzed by food register logs and self-reports.

	V2	V3	*p*-Value of Differences between Visits *p*-Value
Energy intake (kcal) Adherence (%)	104.7 ± 26.2	95.4 ± 23.5	0.02
Carbohydrate intake (grams) Adherence (%)	126.8 ± 43.1	112.4 ± 35.5	0.08
Protein intake (grams) Adherence (%)	104.8 ± 30.3	96.6 ± 29.4	0.06
Fat intake (grams) Adherence (%)	100.3 ± 30.9	65.1 ± 19.6	< 0.001
General Adherence (%) per day	108.6 ± 31.3	93.0 ± 22.1	< 0.001
Patient Adherence § Good 80–110% Bad <80% or >110%	43 (40.5%) 63 (59.4%)	57 (53.8%) 49 (46.2%)	0.56
Self-reported adherence (%) * Telephone visit 1	60 (40–80)	70 (70–80)	< 0.001

* Wilcoxon test, § Chi2 test.

**Table 5 nutrients-13-01744-t005:** Factors associated with good adherence (80–110%) to a nutritional treatment.

Factors	B Coefficient	S.E (Standard Error)	Wald	*p*-Value	Exp (B)	CI (95%)
Female Sex	0.97	0.47	4.15	0.04	2.64	1.03–6.74
Smoking	1.11	0.53	4.46	0.03	3.06	0.11–0.92
Prescribed >1500 kcal	1.88	0.64	8.50	0.004	6.56	0.04–0.54
No Barriers at V3	−3.18	1.07	8.77	0.084	4.59	0.81–25.9

R^2^ = 0.18, Model *p*-value 0.04.

**Table 6 nutrients-13-01744-t006:** Differences before and after the structured intervention.

Characteristic	Baseline	Visit 2	Visit 3	*p*-Value
Body Weight (kg) * § τ	72.1 ± 15.4	71.1 ± 15.4	71.2 ± 15.4	0.005
BMI (Kg/m^2^)	27.1 ± 4.2	26.9 ± 4.6	26.9 ± 4.6	0.29
Systolic blood pressure (mm/Hg)	120 (110–130)	120 (110–120)	120 (110–132)	0.24
Diastolic blood pressure (mm/Hg)	80 (70–80)	80 (78–80)	78 (70–80)	0.17
Waist (cm) * § τ	92.6 ± 12.6	91.5 ± 12.1	90.8 ± 11.9	<0.001
Energy intake (kcal) * § τ	1912 ± 645.1	1722.2 ± 496.7	1547.6 ± 338.7	<0.001
Carbohydrate intake (grams) * § τ	266.6 ± 113.7	228.3 ± 85.3	193.9 ± 53.5	<0.001
Protein intake (grams) * § τ	88.9 ± 32.8	86.4 ± 27.8	80.6 ± 23.6	<0.001
Fat intake (grams) §	63.8 ± 27.9	64.0 ± 20.6	41.3 ± 10.8	<0.001
Fiber (grams/day) * §	29.1 ± 12.1	26.4 ± 7.9	23.4 ± 6.8	<0.001
Physical activity > 150 min/week YES/NO	48 (45.2%)	57 (53.3%)	49 (46.2%)	0.46
Minutes/visit * § τ	60 (50–60)	35 (35–35)	30 (30–30)	<0.001
Glucose (mg/dL)	102.0 ± 23.9	108.3 ± 48.0	102.0 ± 28.6	0.51
Serum Creatinine (mg/dL)	0.85 ± 0.34	0.85 ± 0.37	0.88 ± 0.36	0.65
Triglycerides (mg/dL) * § τ	215.0 (125.7–293)	179.5 (127.5–280)	165.5 (113.5–235)	<0.001
Goal Triglycerides (<150 mg/dL) * § τ	34 (32.1%)	40 (37.7%)	44 (41.5%)	<0.001
Total Cholesterol (mg/dL) * § τ	215.2 ± 56.7	202.7 ± 56.7	198.4 ± 49.7	<0.001
Goal Total Cholesterol (<200 mg/dL) * §	44 (41.5%)	45 (42.4%)	54 (51%)	<0.001
HDL- Cholesterol (mg/dL) § τ	43.6 ± 13.6	42.1 ± 13.6	44.4 ± 14.4	0.09
Goal HDL-Cholesterol (>40 mg/dL)	(58.2%)	(55.2%)	(57.5%)	0.008
LDL-Cholesterol (mg/dL)	135.5 ± 59.5	125.2 ± 46.9	124.7 ± 53.6	0.59
Patients with diabetes (*n* = 21) HbA1c (%)	7.1 ± 1.9	7.5 ± 2.1	7.5 ± 1.7	0.75

Notes: * Significant differences between visit 1 and visit 2; *p* < 0.05 by post hoc, by repeated measures ANOVA or Wilcoxon. § Significant differences between visit 2 and visit 3; *p* < 0.05 by repeated measures ANOVA or Wilcoxon. τ Significant differences between visit 1 and visit 3; *p* < 0.05 by repeated measures ANOVA or Wilcoxon.

## Data Availability

The data presented in this study are available on request from the corresponding author. Due to confidentiality concerns, the data are not publicly available.

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
