# Peer review of "Primary Barriers of Adherence to a Structured Nutritional Intervention in Patients with Dyslipidemia"

_nutrients, 2021, doi:10.3390/nu13061744_

Round 1

Reviewer 1 Report

 Identification of the primary barriers patients with dyslipidemias face adhering to a structured nutritional intervention

The authors aim to identify the barriers to adherence for MNT for dyslipidemia using a structured nutrition intervention. It also seeks to evaluate the effect of the intervention on adherence and lipid profile. The document is important for researchers in the field. It has good summary tables and a sound introduction of the need for this work, and a well-structured discussion. However, it needs some clarifications in methodology and interpretation of the results.   

Cost, lack of time, energy, and accessibility are commonly perceived barriers to making dietary changes documented in many settings. However, evaluation of knowledge, skills, and abilities –including self-efficacy and stages of change–  of participants before and after the intervention can present a better picture of the results of this MNT intervention. The lack of evaluation of knowledge or motivation –stages of change-, and measuring adherence with a 3d-diet recall are recognized limitations of this study. The degree of dietary adherence required to observe changes in health outcomes is difficult to evaluate objectively by self-reported dietary records. Still, the authors point out the need to educate all team members in recognizing the need to implement tailored interventions and empathize the importance of nutrition education. In addition, the measurement of dietary biomarkers can provide an objective profile of the level of adherence to MNT. Indeed, more work is required in this area.

Major

Self-reported dietary information has many limitations. There is a need to develop and report more objective methods of identifying dietary patterns to help reduce measurement error. The authors measure changes in dietary patterns through self-reported measures such as 24-hour recalls, food frequency questionnaires, and food logs. These methods serve as cost-effective tools to identify dietary patterns, food intake, and food behaviors that cannot be determined through more objective measures. However, there is a significant measurement of error when self-reported measures such as 24-hour recalls or 3-day food logs to calculate energy intake. This error is due to the underreporting of energy-dense foods and overall energy intake, which is a major concern, especially among individuals affected by overweight or obesity (i.e., Subar, A.F., J Nutr 2015;145: 2639-2645). What is the estimated reported error for this population? Do the authors believe that one 24-hr food recall is enough to prescribe the nutrition plan? Where is the evidence supporting this? In what proportion of cases the 24-hr food recall was used instead of the 3-d food recall?

Is the questionnaire used to identify barriers to adherence standardized or validated? Are there any references for the revised literature (1st paragraph, section 2.1)?

Was the goal to solve adherence barriers or attachment barriers (2nd paragraph, section 2.1)? Please correct punctuation marks.

The sentence “Trying also to provide the tools to eliminate or reduce the detected barrier” seems to be incomplete

When the authors noted, “We implemented two different (dietary?) plans depending on the needs of the patient,” do they refer to energy needs or outcome goals?

The recommendation from ATP IV clinical guidelines to not specifiy the type and amount of dietary fat recommended beyond 30 to 40%. Was any advice regarding the replacement of saturated fat with monounsaturated or PUFA’s?

In the multivariate analysis. For the second criteria of a previous statistically significant correlation (p<0.005), is that correct? How many correlations were in that category? What variables had a significant p-value (<0.05) or biological plausibility to remain in the final model? For what outcome?

The authors define the criteria for “good” and “bad” adherence in the methods section. However, in the results, they note, “Fat intake adherence was excellent between V1 to V2 (100.3%); however, it declined significantly to 64.1% (p <0.001), during V3, and was therefore considered poor”. What is excellent adherence?

Section 3.2. It is not clear why the authors decide to note “however, even though this variable does not have a statistically significant p-value, it tends to be significant”? (incidentally, for Table 5, please format to have the prescribed intake [>1500 kcal] in the second line).

Minor

Please use the institutional acronym, INCMNSZ, with the authorship for clarification.

Be consistent in the use of MNT abbreviation, and sometimes (p. 2 L. 16, section 3.1) not used.

Please spell the word three instead of using a number a the beginning of the sentence (1st paragraph, section 2.1)?

Please correct figure 1 and hide paragraph marks

Discussion Section, please revise the position of the references. Some of them are referred inconsistently to the information discussed.

References: There is no consistent format of the references. The names of journals are abbreviated at least in three different styles. Examples include references 1, 11,13,23,26.

Reviewer 2 Report

The authors present a very interesting study addressing the barriers for structured nutritional interventions in patients with dyslipidemia. These barriers may also be considered for other groups of patients. 

 Some comments:

-The title may be simplified: Primary barriers of adherence to a structured nutritional intervention in patients with dyslipidemia.

-There is a potential methodological bias by two difffernt nutritional plans and food-recall periods. This should be discussed and added to the limitations of the study .

-Please specify: How were the 3-day and  24hr-food recall analyzed regarding macronutrient and energy intake? - Software?

-The difference between self-reported and measured adherence may be even more clarified. 

-Please comment on the decreasing protein and fat adherence between V2 and V3.  
